# Trust development as an expectancy-learning process: Testing contingency effects

**Guy Bosmans**[1]*, **Theodore E. A. Waters**[2], **Chloe Finet**[1,2], **Simon De Winter**[1], **Dirk Hermans**[3]

1 Clinical Psychology, KU Leuven, Belgium, 2 New York University - Abu Dhabi, United Arab Emirates, 3 Centre for Psychology of Learning and Experimental Psychopathology, KU Leuven, Belgium

* guy.bosmans@kuleuven.be

**Data Availability Statement:** Data can be found at https://doi.org/10.6084/m9.figshare.9777722.v1.

**Funding:** This research was supported by FWO grants G077415N and G075718N, awarded to GB. The funders had no role in study design, data

## Abstract

Trust in parental support and subsequent support seeking behavior, a hallmark of secure attachment, result from experiences with sensitive parents during distress. However, the underlying developmental mechanism remains unclear. We tested the hypothesis that trust is the result of an expectancy-learning process condtional upon contingency (the probability that caregiver support has a positive outcome). We developed a new paradigm in which a novel caregiver provides help to solve a problem. Contingency of the caregiver's support was manipulated and participants' trust in the caregiver and their help seeking behavior was measured in three independent samples. The hypothesis was supported suggesting that trust and support seeking result from an expectancy-learning process. These findings' potential contribution to attachment theory is discussed.

## Introduction

The extent to which children are able to subjectively trust in primary caregivers as a resource to rely upon when experiencing distress has an important impact on developmental outcomes in several domains such as psychopathology, academic success, social competence, and general health [1–3]. Consequently, understanding how trust develops has traditionally been of critical interest. Trust development has been most consistently studied in the context of attachment research that considers trust as a hallmark of secure attachment [4]. It has been suggested that trust develops when caregivers are consistently sensitive and responsive [5, 6]. However, this research has been limited to cross-sectional and longitudinal studies testing the correlations between broadband measures of sensitive parenting and attachment. As a result, little is known about the specific mechanisms that explain trust development in day to day or moment to moment interactions. Broadband studies on parenting and attachment have been unable to explain the majority of the variance in attachment [6], for which little robust explanation exists to date [7, 8]. Therefore, the current study aimed to experimentally investigate the development of subjective trust at a more micro-process level.

For this purpose, we developed a new research paradigm based on Waters and Waters' [9] recent insight that subjective trust is an expectation resulting from a cognitive learning process. Learning research has shown that expectations about neutral stimuli (conditional

collection and analysis, decision to publish, or
preparation of the manuscript.

**Competing interests:** The authors have declared
that no competing interests exist.

stimulus, CS) change when the CS gets associated with the occurrence of a second, meaningful stimulus (unconditional stimuli, UCS) that automatically elicits an emotionally relevant response (unconditional response, UCR). Changes in the meaning of the CS can be observed in terms of the acquired expectation (conditional reaction, CR) that the CS will elicit the UCR. This is a classical conditioning learning process [10] which has also been described as an expectancy-learning process [11–13]. Once the CS elicits certain behavioral responses, those responses may get reinforced through a process of operant conditioning [14]. A discriminative stimulus (Sd) increases the likelihood of certain behaviors (R) that have in the past resulted in positive effects, which further reinforce the behavior (Sr).

The idea that learning theory could explain at least a part of attachment development, has traditionally been focus of fierce debate [15]. When developing his attachment theory, Bowlby was in a difficult position. On the one hand he wanted to emphasise that attachment is an evolutionarily primed behavior system, not reducible to classical or operant conditioning [16]. On the other hand, he also held that the attachment behavioral system is assembled and elaborated in the context of learned experiences [17]. Nevertheless, he mostly left the topic alone. Later, attachment researchers mostly demonstrated that children develop their attachment relationships independent of the quality of care, interpreting this finding as evidence that the attachment behavioral system is an evolutionary driven system that requires no learning experiences to be established [15, 18].

More recently, researchers started to argue that this biological preparedness of infants to establish attachment relationships with caregivers does not fully explain why individual differences in children's expectations or trust about caregivers' availability for support develop [8]. One thus far understudied possibility could be that learning models might be a helpful addition to attachment theory to explain attachment-related differences that are not innate [19], such as secondary (anxious, avoidant, and secure) attachment styles [20]. Recent literature argues in a similar way that adult attachment development reflects a conditioning process [21–23]. Focusing on childhood attachment, Bosmans [24] proposed that children learn to trust (CR) in a responsive primary caregiver like the mother (CS) when the mother gets associated with the repeated experience that she provides successful support during distress (UCS), which is automatically followed by a sense of relief and a sense of security (UCR). At the level of operant conditioning, it has been proposed that trust versus lack of trust in support during distress (Sd) increases or decreases the likelihood that children will seek support (R) during distress. More versus less support seeking is respectively reinforced by the fact that sensitive caregivers help to solve problems more easily or by the fact children avoid experiencing the anticipated negative effects of rejection or inadequate support [24]. In sum, the current study builds on the idea that the processes of the attachment system are innate, but that attachment styles are (at least partly) shaped by learning.

One advantage of applying learning theory to subjective trust development, is that it allows formulating very specific research questions and testing concrete hypotheses about the mechanisms underlying trust development. In the current study, we focused specifically on the role of contingency. Contingency is considered a central mechanism in conditioning that refers to the relative probability that an US occurs in the presence of a CS [13, 25]. One straightforward prediction drawn from learning theory, is that higher contingency between a CS and a positive UCS is linked to a more positive expectation about that stimulus. Applied to the development of subjective trust, contingency would refer to the percentage of the occasions (or single learning events) during which children have a problem in which mothers help to solve that problem.

The idea of contingency can be linked to the concept of sensitive caregiving. Sensitivity refers to a caregiver's ability to detect children's needs and to respond to them promptly and

adequately [26, 27]. Consistent sensitive care has robustly been shown to be an important predictor of trust development [5]. However, there is a general consensus that qualitative differences in sensitive parenting do not organize along good versus bad caregiving categories. Instead it has been theoretically argued that parents need to be "good enough" [28]. Although the concept of good enough mothering has been generally accepted, little is known *when* or *how much* sensitive parenting is good enough to significantly increase subjective trust and to stimulate help seeking behavior. In addition, little is known about how changes in sensitive parenting over time (from more to less sensitive parenting and vice versa) affect children's trust development. These are questions that can be approached from a conditioning perspective and that can be further tested by manipulating the contingency or success of a caregiver's help.

The current study aimed to provide proof of concept for the idea that contingency could explain part of the variance in the development of trust (Research Question 1) and in the development of support seeking (Research Question 2) and for the idea that questions regarding the definition of good enough mothering and regarding the dynamics of (in)stability of trust over time can in theory be explained by contingency-related effects (Research Question 3). To this aim, we developed a new paradigm in which participants are introduced to a new caregiver and in which participants have to solve a challenging task. The paradigm consists of two phases. In the learning phase, participants always get advice from the new caregiver to help solve the task. In the test phase, participants need to decide whether or not to ask the caregiver for help. During each phase, we can manipulate the contingency of caregiver support, ranging from 100% (the caregiver is always helpful) to 0% (the caregiver is always unhelpful). A 50% contingent caregiver is unpredictable (half of the time the caregiver is helpful; half of the time the caregiver is not helpful). To test the effect of the contingency manipulation, at baseline, after the learning phase, and after the test phase, we can measure participants' trust in the novel caregiver's support.

Although this approach will not straightforwardly translate to the complex interactions with a real attachment figure during critical timepoints in individuals' lives, the approach has the advantage that we can identify the role of contingency controlling for the impact of prior learning experiences with those real attachment figures. This way, we can experimentally test whether differences in contingency affect trust and test how changes in contingency from the learning to the test phase are linked to changes in trust. Finally, we can measure participants' inclination to seek the caregiver's help during the test phase, as a proxy of participants' support seeking behavior.

In sum, in three studies we investigated the following questions: (1) does higher contingency in terms of successful support by a caregiver lead to increased trust, and (2) to more help seeking behavior, and (3) does change in contingency affect trust and help seeking behavior.

## Study 1

We first piloted the paradigm in adults. To test research question 1 that higher contingency is linked with higher trust, participants were randomly assigned to either a 100%, 70%, or a 50% contingent caregiver in the learning phase. This manipulation additionally allowed to test whether 70% learning phase contingency was already "good enough" to promote trust. To test research question 2 that higher contingency affects help seeking behavior, we looked at the first five trials of the test phase to see whether higher learning phase was linked to more help-seeking behavior. To test research question 3 that change in contingency has subsequent effect

on trust and help seeking behavior, in the test phase, if participants chose to ask the caregiver's help, contingency was 100%.

## Participants and procedure

In this pilot study, 30 adults participated (*M*age = 28.53; *SD*age = 8.05; 21 males). Adults were randomly assigned to the conditions: 8 in the 50% contingency condition, 7 in the 70% contingency condition, and 6 in the 100% contingency condition. Participants were fully informed about the goal and content of the experiment and only participated after they gave written consent. This procedure was approved by the Social and Societaly Ethics Commission of the KU Leuven (Belgium).

To manipulate contingency, we designed a challenging task during which participants get help from a caregiver. The task was based on the hungry donkey task [29], which is often used in the study of hot cognitive processes. Thus, the task is known to elicit sufficient levels of distress [30]. During this task, participants have to seek food (apples) to feed a hungry donkey. In the current paradigm, apples are hidden behind four closed doors (Fig 1b). Behind each door, a predetermined number of green or red apples were present. During each trial, one door hid five green apples, one door hid one green apple, one door hid five red apples, and one door hid one red apple (Fig 1c). The position of these apples (i.e., the door behind which the green and red apples were hidden) was randomized across trials. When participants chose a door hiding green apples, the number of green apples was added to their score. In contrast, when a door hiding red apples was chosen, the number of red apples was subtracted from their score.

We constructed two phases in the hungry donkey task. First, we designed a learning phase. We introduced an unfamiliar caregiver, an avatar, that gave advice on which door to open. Participants were free to follow the advice or not. Once a door was chosen, all doors were opened to show how the apples were distributed during that specific trial (Fig 1c). A total score was registered in the upper left corner of the screen and participants' total score was tracked on a separate scale (the hunger scale). Once participants reached 20 points on the hunger scale, a screen thanked participants for feeding the donkey stimulating them to continue in the same manner (Fig 1d). The learning phase consistend of 10 trials, based on research showing that this is the adequate number of trials to manipulate contingency in experimental designs [31]. Afterwards, the hunger meter was reset to zero and participants recommenced. This learning phase allowed us to manipulate contingency in terms of likelihood that the advice of the caregiver was helpful. Participants were assigned to one of three contingency conditions: 50%, 70%, and 100% contingency. The order in which the caregiver presented correct or incorrect advice was randomized. When the advice was correct, the caregiver suggested the door hiding five green apples. When the advice was incorrect, the caregiver would suggest the door hiding five red apples. After completing the learning trials, participants were asked to rate their level of trust in the caregiver (Fig 1a).

Second, we designed a test phase, consisting of 20 trials. During this phase, the caregiver was not automatically present. In contrast, participants had to decide whether to call the caregiver using a button in the center of the screen (Fig 1e). Before the start of the trials, participants were reminded this caregiver would provide advice on which door to choose, but also that this advice could be incorrect. They were instructed that they could choose whether to call the caregiver, and whether they would follow the caregiver's advice (Fig 1f). The caregiver's appearance was identical to the learning phase. However, in contrast to the learning phase and unbeknownst to the participants, the caregiver in the first testing phase was correct 100% of the trials, regardless of the participants' assigned learning phase condition. Again, after completing the trials, participants were asked to rate their level of trust in the caregiver (Fig 1a).

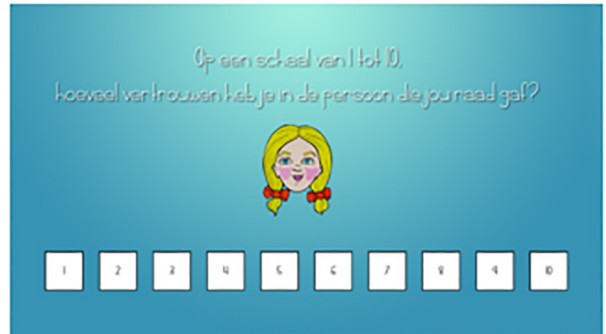

a = trust measure

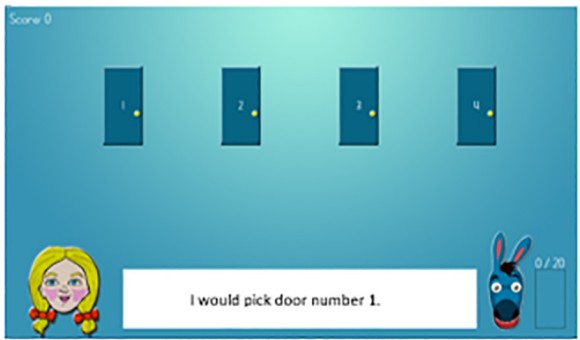

b = the caregiver gives advice

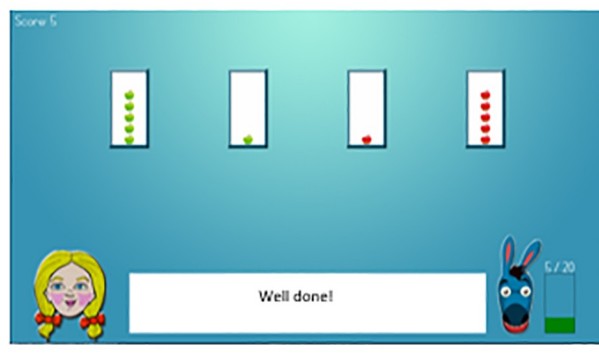

c = the result of the child's choice

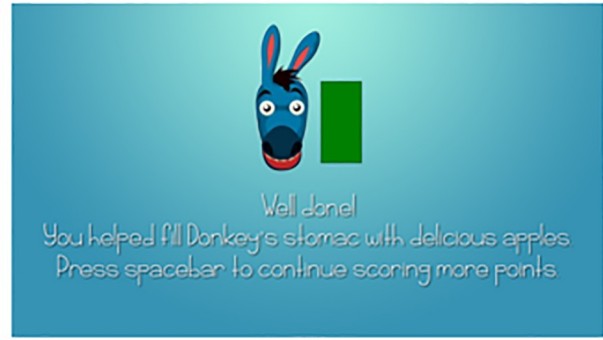

d = congratulation screen

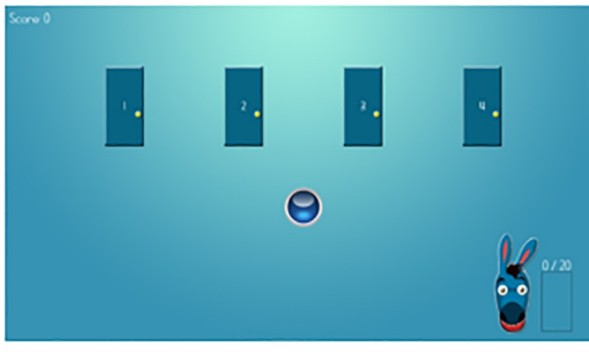

e = button to call for the caregiver's help

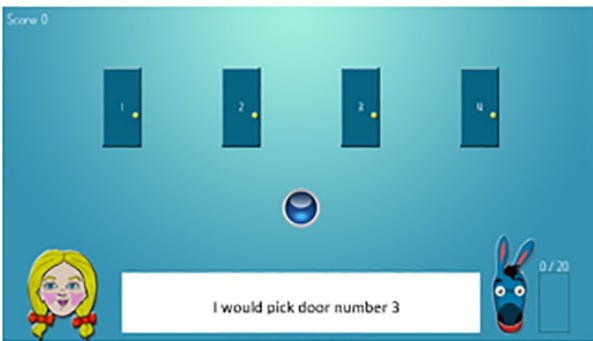

f = the caregiver's suggestion

**Fig 1. Trial presentation.**

## Results and discussion

**Is higher contingency related to increased trust?.** We conducted a one-way ANOVA with condition (50%, 70%, or 100% learning phase contingency) as independent variable and post learning phase trust as dependent variable, followed by Bonferroni post-hoc tests. Table 1 shows that after the learning phase, levels of trust were significantly higher in the higher learning phase contingency conditions (70 vs 50: mean difference = 2.60, 95% CI = [0.69,

**Table 1. Means, Standarddeviations, One-Way ANOVA's and paired-samples *t*-tests for Study 1.**

| | Trust 1 | | Trust 2 | | % of trials in which participant asked for help | | | | | | | | | |
| | | | | | Trials 1–5 | | Trials 6–10 | | Trials 11–15 | | Trials 16–20 | | Trials Total | |
| | *M* | *SD* | *M* | *SD* | *M* | *SD* | *M* | *SD* | *M* | *SD* | *M* | *SD* | *M* | *SD* |
|---|---|---|---|---|---|---|---|---|---|---|---|---|---|---|
| 50 | 2.90 | 1.85 | 4.90 | 3.87 | 1.10 | 1.45 | 2.60 | 2.12 [b] | 2.20 | 2.20 | 2.50 | 2.27 | 8.40 | 7.38 |
| 70 | 5.50 | 2.17 | 8.00 | 2.16 [a] | 2.90 | 1.97 | 3.80 | 1.75 [b] | 3.70 | 1.83 | 3.80 | 1.81 | 14.20 | 6.89 |
| 100 | 9.60 | 0.52 | 9.80 | 0.42 | 4.40 | 1.08 | 4.60 | 1.26 | 4.10 | 1.73 | 4.40 | 1.27 | 17.50 | 4.48 |
| F(2,27) | 40.64*** | | 9.29*** | | 11.48*** | | 3.32, *p* = .051 | | 2.69, *p* = .086 | | 2.81, *p* = .078 | | 5.22* | |
| Post-hoc | 50 < 70 < 100 | | 50 < 70 < 100 | | 50 < 70 = 100 | | 50 = 70 = 100 <br> 50 < 100 | | 50 = 70 = 100 | | 50 = 70 = 100 | | 50 = 70 = 100 <br> 50 < 100 | |

Trust 1 = Trust after the Learning phase; Trust 2 = Trust after the test phase. Paired samples *t*-tests within condition comparing adjacent measurement points:

[a] *p* < .01,

[b] *p* < .05

*** *p* < .001;

* *p* < .05.

4.51], *p* < .01, Cohen's *d* = 1.29; 100 vs 50: mean difference = 6.70, 95% CI = [4.79, 8.61], *p* < .001, Cohen's *d* = 4.93; 100 vs 70: mean difference = 4.10, 95% CI = [2.19, 6.01], *p* < .001, Cohen's *d* = 2.60). The significant difference between the 70% and 100% contingency conditions suggested that 70% contingency was not yet good enough to increase trust.

**Is higher contingency related to more help seeking behavior?.** We conducted a one-way ANOVA with condition (50%, 70%, or 100% learning phase contingency) as independent variable and help seeking in the first five test phase trials as dependent variable, followed by Bonferroni post-hoc tests. Table 1 shows that the manipulation had an effect on the total amount of times participants pressed the button to call the caregiver during the first five test trials. Participants from the 50% learning phase condition sought significantly less help than participants from the 70% and 100% learning phase (70 vs 50: mean difference = 1.80, 95% CI = [0.04, 3.56], *p* < .05, Cohen's *d* = 1.04; 100 vs 50: mean difference = 3.30, 95% CI = [1.54, 5.06], *p* < .001, Cohen's *d* = 2.58). Participants from the 70% and 100% learning phase conditions were as likely to seek help in the test phase. (100 vs 70: mean difference = 1.50, 95% CI = [-0.26, 3.26], *p* = .116, Cohen's *d* = 0.94).

**Does change in contingency affect trust and help seeking behavior?.** First, with regard to trust, we conducted a one-way ANOVA with condition (50%, 70%, or 100% learning phase contingency) as independent variable and trust after the test phase as dependent variable, followed by Bonferroni post-hoc tests. Table 1 shows that the initial learning phase condition effects remained significant with lower contingency being related to lower levels of trust (70 vs 50: mean difference = 3.10, 95% CI = [0.17, 6.04], *p* = .036, Cohen's *d* = 0.99; 100 vs 50: mean difference = 4.90, 95% CI = [1.97, 7.84], *p* < .001, Cohen's *d* = 1.78; 100 vs 70: mean difference = 1.80, 95% CI = [-1.14, 4.74], *p* = .387, Cohen's *d* = 1.16). Moreover, paired samples t-tests within each learning phase condition showed that only the 70% learning phase condition participants' trust significantly improved after experiencing a 100% contingent caregiver in the test phase (mean difference = -2.50, 95% CI = [-3.63, -1.37], *t*(9) = -5.00, *p* < .001, Cohen's *d* = -1.15). This catch-up resulted in levels of trust comparable to the 100% learning condition participants. Instead, the 50% learning condition participants did not significantly benefit from the altered learning experiences (mean difference = -2.00, 95% CI = [-5.07, 1.07], *t*(9) = -1.47, *p* = .175, Cohen's *d* = 0.65). This suggests in adult participants that initial contingency effects had a lasting influence in trust development, even after exposure to a novel, improved experience.

Second, with regard to help seeking behavior, we conducted a one-way ANOVA with learning phase condition as independent variable and calling behavior in each of the three last blocks of test trials and the total trials. Table 1 shows that the learning phase condition effect did not fully reach significance during all subsequent blocks of five trials. Nevertheless, across all 20 trials, the initial learning phase manipulation retained its effect (70 vs 50: mean difference = 5.80, 95% CI = [-1.48, 13.08], $p$ = .156, Cohen's $d$ = 0.81; 100 vs 50: mean difference = 9.10, 95% CI = [1.82, 16.38], $p$ = .011, Cohen's $d$ = 1.49; 100 vs 70: mean difference = 3.30, 95% CI = [-3.98, 10.58], $p$ = .772, Cohen's $d$ = 0.57).. There was a significant increase in help seeking behavior from block 1 to block 2 for the 50% (mean difference = -1.50, 95% CI = [-2.68, -0.32], $t(9)$ = -2.88, $p < .05$, Cohen's $d$ = -0.83) and 70% (mean difference = -0.90, 95% CI = [-1.76, -0.04], $t(9)$ = -2.34, $p < .05$, Cohen's $d$ = -0.48) learning phase participants, but in the subsequent blocks their help seeking frequency remained stable. As a result, especially the 50% learning phase participants continued to lag behind the 100% learning phase children with regards to help seeking.

A limitation of the study was that we only measured trust after the manipulation. With a baseline measure, it would be easier to draw conclusions about the causal role of contingency in trust development. Moreover, evidence for contingency effects was needed in younger participants to be more certain that contingency could be a relevant mechanism to understand children's trust development. Study 2 aimed to solve these limitations.

## Study 2

The setup of the second study was largely identical to Study 1, except for three aspects. First, we included a baseline measure of trust in the novel caregiver. Second, the study was conducted in middle childhood. Research suggests that this age-period is characterized by biological changes that facilitate trust learning [32, 33]. Consequently, to evaluate whether contingency is relevant for trust development, contingency effects should also be found in middle childhood. Third, we also administered a measure of children's trust in their mother. Theory suggests that relational experiences with care during distress gradually internalize into a blueprint on which children evaluate first encounters in new relationships [9]. This has for example been evidenced for children's relationships with teachers [34]. To evaluate the theoretical relevance of the trust in the paradigm's avatar, we tested the correlation between children's trust in their mother's support and children's trust in the paradigm's caregiver at baseline. During the learning phase, Study 1's 50-70-100% contingency conditions were retained. During the test phase, the caregiver was again 100% contingent.

### Participants and procedure

The sample (N = 65) consisted of 32 girls (49.2%) and 33 boys (50.8%) between the age of 9 and 12 years old (M = 10.22, SD = 1.05). All children in the sample were Caucasian. Of these children, 55 lived with both biological parents (84.6%), 9 children had parents who were divorced (13.6%), one child's mother was deceased (1.5%), and one child did not respond to the question (1.5%). Excluding one child (1.5%) who did not respond to the question, all children considered their biological mother as their primary attachment figure (98.5%).

Participants were recruited from an elementary school and an out-of-school care facility where children can spend their time while their parents are off to work during summer holiday. Parents were informed about the goal of the study beforehand and asked to sign an informed consent form. Children were only invited to participate after we obtained written consent of the parents. Before the experiment took place, all children were informed about the goals of the study. Children only participated if they gave their written consent. Children were

tested in groups. Children were instructed not to talk during the experiment. They were randomly assigned to the three conditions (50% = 22; 70% = 22; 100% = 21). This procedure was approved by the Social and Societaly Ethics Commission of the KU Leuven (Belgium).

## Method

Children's trust-related expectations of maternal support were assessed using the People In My Life Questionnaire's Trust-subscale (PIML) [35]. Expectations of trust are positive affective/cognitive experiences of trust in the accessibility and responsiveness of the attachment figure (10 items; e.g., "I can count on my mother to help me when I have a problem"). Children are asked to respond to these questions using a 4-point Likert scale, ranging from 1 ("almost never true") to 4 ("almost always true"). Research has demonstrated that the trust-subscale is associated with support seeking behavior [36], thus confirming its validity. Higher scores on the Trust-subscale indicate more trust in maternal support. Internal consistency was good ($\alpha$ = .86).

## Results and discussion

**Is higher contingency related to increased trust?.** First, we found a small correlation between baseline trust in the novel caregiver and children's trust in maternal support, $r(62)$ = .24, $p$ = .062. This suggested that the trust studied in the current experimental paradigm might be at least somewhat relevant to understand the development of trust in primary caregivers. Second, we conducted a 2 (Within Subjects: Time: Baseline, Post learning Phase) X 3 (Between Subjects: Condition: 50%, 70%, and 100% learning phase contingency) repeated measures ANOVA on trust. This revealed a significant Time X Condition interaction effect during the learning phase, $F(2, 60)$ = 28,50, $p$ < .001, $\eta_p^2$ = 49. Table 2 shows that contingency differences caused significant differences between all three conditions with higher contingency being linked to more trust after the learning phase (70 vs 50: mean difference = 1.36, 95% CI = [0.001, 2.73], $p$ < .05, Cohen's $d$ = 0.66; 100 vs 50: mean difference = 5.00, 95% CI = [3.62, 6.38], $p$ < .001, Cohen's $d$ = 3.43; 100 vs 70: mean difference = 3.64, 95% CI = [2.26, 5.02],

**Table 2. Means, Standarddeviations, One-Way ANOVA's and paired-samples *t*-tests for Study 2.**

| | Trust 1 | | Trust 2 | | Trust 3 | | % of trials in which participant asked for help | | | | | | | | | |
|---|---|---|---|---|---|---|---|---|---|---|---|---|---|---|---|---|
| | | | | | | | Trials 1–5 | | Trials 6–10 | | Trials 11–15 | | Trials 16–20 | | Trials Total | |
| | *M* | *SD* | *M* | *SD* | *M* | *SD* | *M* | *SD* | *M* | *SD* | *M* | *SD* | *M* | *SD* | *M* | *SD* |
| 50 | 6.36 | 1.50 | 4.00 | 1.69[a] | 6.73 | 2.62[a] | 2.91 | 1.60 | 3.14 | 1.86 | 3.41 | 1.99 | 3.36 | 2.06 | 12.82 | 6.74 |
| 70 | 6.76 | 1.79 | 5.36 | 2.40[b] | 8.00 | 1.69[a] | 3.41 | 1.53 | 3.91 | 1.57[b] | 3.82 | 1.74 | 4.00 | 1.66 | 15.14 | 5.51 |
| 100 | 6.00 | 1.08 | 9.00 | 1.18[a] | 9.19 | 2.04 | 4.19 | 1.21 | 4.24 | 1.34 | 4.10 | 1.37 | 4.10 | 1.61 | 16.92 | 4.49 |
| F(2,64) | 1.35 | | 42.33*** | | 7.03** | | 4.19* | | 2.68+ | | .86 | | 1.07 | | 2.46+ | |
| Post-hoc | 50 = 70 = 100 | | 50 < 70 < 100 | | 50 = 70 = 100 50 < 100 | | 50 = 70 = 100 50 < 100 | | 50 = 70 = 100 50 < 100+ | | 50 = 70 = 100 | | 50 = 70 = 100 | | 50 = 70 = 100 50 < 100+ | |

Trust 1 = Trust at Baseline, due to a technical error data was not recorded for this measure, so for this analysis, the degrees of freedom were 2 and 62; Trust 2 = Trust after the Learning phase; Trust 3 = Trust after the test phase. Paired samples *t*-tests within condition comparing adjacent measurement points:

[a] p < .001,

[b] p < .10

*** *p* < .001;

** *p* < .01;

* *p* < .05;

+ *p* < .1;

$p < .001$, Cohen's $d = 1.92$). Moreover, paired-samples t-tests showed in the 50% contingency condition a significant drop in trust during the learning phase (mean difference = 2.36, 95% CI = [1.34, -3.38], $t(21) = 4.82$, $p < .001$, Cohen's $d = 1.48$). In the 70% condition, the manipulation did not lead to an increase in trust during the learning phase. At best, there seemed to have been a trend towards a drop in trust during the learning phase (70% learning phase: mean difference = 1.38, 95% CI = [-0.10, 2.86], $t(20) = 1.95$, $p = .066$, Cohen's $d = 0.64$). In the 100% condition, there was a significant increase in trust during the learning phase (mean difference = -3.10, 95% CI = [-3.83, -2.37], $t(19) = -8.93$, $p < .001$, Cohen's $d = -2.82$). Like in Study 1, this suggests that trust development depended on the contingency of success in provided care and that trust development in the current paradigm required more than 70% contingent caregivers.

**Is higher contingency related to more help seeking behavior?.** We conducted a one-way ANOVA with condition (50%, 70%, or 100% learning phase contingency) as independent variable and help seeking in the first five test phase trials as dependent variable, followed by Bonferroni post-hoc tests. Table 2 shows that children in the 50% learning phase condition called significantly less for help compared to the 100% learning phase condition children (mean difference = 1.28, 95% CI = [0.18, 2.38], $p < .05$, Cohen's $d = 0.90$). Like in Study 1, the 70% learning phase condition children displayed the same level of help seeking behavior as the 100% children (mean difference = 0.78, 95% CI = [-0.32, 1.88], $p = .254$, Cohen's $d = 0.57$).

**Does change in contingency affect trust and help seeking behavior?.** First, with regard to trust, a 2 (Within Subjects: Time: Baseline, Post learning Phase) X 3 (Between Subjects: Condition: 50%, 70%, and 100% learning phase contingency) repeated measures ANOVA on trust showed a significant Time X Condition interaction effect during the test phase, $F(2, 60) = 28,50$, $p < .001$, $\eta_p^2 = 49$. To probe this effect, we conducted a one-way ANOVA with condition (50%, 70%, or 100% learning phase contingency) as independent variable and trust after the test phase as dependent variable, followed by Bonferroni post-hoc tests. Moreover, we conducted paired samples t-tests within each learning phase condition to test whether trust changed over time. Table 2 shows that, after the 100% contingent test phase, trust increased in both the 50% and 70% learning phase conditions (50% learning phase: mean difference = -2.73, 95% CI = [-3.91, -1.55], $t(21) = -4.81$, $p < .001$, Cohen's $d = -1.24$; 70% learning phase: mean difference = -2.64, 95% CI = [-3.68, -1.59], $t(21) = -5.24$, $p < .001$, Cohen's $d = -1.27$). However, while ANOVA showed that the 70% condition statistically caught up with the 100% condition (100 vs 70: mean difference = 1.19, 95% CI = [-0.43, 2.81], $p = .225$, Cohen's $d = 0.64$), the 50% condition children still trusted the caregiver significantly less than the 100% condition children mean difference = 2.46, 95% CI = [0.85, 4.08], $p < .01$, Cohen's $d = 1.05$). Like in Study 1, this suggests that improving contingency increased trust, but that prior less contingent care experiences delayed this catch-up.

With regard to help seeking behavior, a one-way ANOVA with learning phase condition as independent variable and calling behavior in each of the three last blocks of test trials and the total trials, showed that the learning phase condition effect was largely erased during trials 6–10 (block 2: 70 vs 50: mean difference = 0.77, 95% CI = [-0.42, 1.97], $p = .348$, Cohen's $d = 0.45$; 100 vs 50: mean difference = 1.10, 95% CI = [-0.11, 2.31], $p = .085$, Cohen's $d = 0.68$; 100 vs 70: mean difference = 0.33, 95% CI = [-0.88, 1.54], $p = 1.00$, Cohen's $d = 0.23$) and was fully erased in the subsequent trials.

## Study 3

Study 3 aimed to test the robustness of the findings in an independent sample of children in middle childhood. Moreover, Study 2 did not reveal which percentage of contingency suffices

to increase trust levels increase compared to baseline. Therefore, we included an 80% instead of a 70% contingent condition, to test whether 80% contingency suffices to significantly increase trust. Additionally, a 50% contingent caregiver is unpredictable, but still provides half of the time successful support. A clinically relevant question is whether children's trust can catch up when having good learning experiences with caregivers after prior predictably negative experiences. Therefore, we included a 20% instead of a 50% learning phase condition, followed by a 100% contingent test phase condition. Finally, we were interested to see whether children who learn that caregiving is 100% contingent are more protected against the negative effects of subsequent negative caregiving experiences. Therefore, we manipulated the contingency in the test-phase adding a 20% contingent condition to this phase.

## Participants and procedure

A total of 170 children participated ($M_{age}$ = 9.58; $SD_{age}$ = 1.49; 49.7% boys). Only 14.6% of the children came from divorced families. All children reported trust in their biological mother, except for one child who answered questions on the step-mother. Age and gender were equally distributed over conditions and did not affect results. All participants were recruited from elementary schools. The recruitment, information, consent and administration procedure was equal to Study 2.

## Method

Trust in maternal support was measured with the PIML. Cronbach's alpha was .78.

## Results and discussion

**Is higher contingency related to increased trust?.** First, baseline trust in the novel caregiver was significantly correlated with trust in maternal support, $r(170)$ = .16, $p$ = .043. Second, we conducted a 2 (Within Subjects: Time: Baseline, Post learning Phase) X 3 (Between Subjects: Condition: 20%, 80%, and 100% contingency) repeated measures ANOVA on trust. This revealed a significant Time X Condition interaction effect, $F(2, 167)$ = 24.13, $p < .001$, $\eta_p^2$ = .22. Table 3 shows that trust significantly decreased in the 20% condition, (20% learning phase: mean difference = 1.65, 95% CI = [0.85, 2.45], $t(70)$ = 4.11, $p < .001$, Cohen's $d$ = 0.67), but significantly increased in both the 80% and 100% conditions (80% learning phase: mean difference = -1.23, 95% CI = [-1.88, -0.58], $t(65)$ = -3.77, $p < .001$, Cohen's $d$ = -0.61; 100% learning phase: mean difference = -2.12, 95% CI = [-3.10, -1.14], $t(32)$ = -4.40, $p < .001$, Cohen's $d$ = -1.02). Moreover, a one-way ANOVA with condition as independent variable and post learning phase trust as dependent variable, showed there was no trust-difference between the 80% and 100% condition (80 vs 100: mean difference = 1.11, 95% CI = [-0.16, 2.38], $p$ = .110, Cohen's $d$ = 0.47). So, this suggests for the current experimental procedure that contingency needs to be at least 80% to be good enough for trust in the novel caregiver to increase.

**Is higher contingency related to more help seeking behavior?.** We conducted one-way ANOVAs with condition (20%, 80%, or 100% learning phase contingency) as independent variable and help seeking in the first five test phase trials as dependent variable, followed by Bonferroni post-hoc tests. Analyses were conducted for the 100% and 20% contingent test phase conditions separately (see Table 3). For the 100%contingent test phase, there was no significant difference in the amount of times children called for help when comparing the 20% and 80% learning phase contingency conditions (80 vs 20: mean difference = -0.57, 95% CI = [-1.54, 0.41], $p$ = .250, Cohen's $d$ = -0.28). This result seems in line with Study 2 where the 50% and 70% learning phase contingency conditions yielded similar levels of help seeking behavior.

**Table 3. Means, Standarddeviations, One-Way ANOVA's and paired-samples *t*-tests for Study 3.**

| | Trust 1 | | Trust 2 | | | Trust 3 | | % of trials in which participant asked for help | | | | | | | | | |
| | | | | | | | | Trials 1–5 | | Trials 6–10 | | Trials 11–15 | | Trials 16–20 | | Trials Total | |
| Learning | M | SD | M | SD | Test | M | SD | M | SD | M | SD | M | SD | M | SD | M | SD |
|---|---|---|---|---|---|---|---|---|---|---|---|---|---|---|---|---|---|
| 20 | 5.31 | 2.21 | 3.66 | 2.72 [a] | 20 | 3.25 | 2.85 | 1.86 | 1.59 | 1.25 | 1.56 [a] | 1.03 | 1.48 | 1.33 | 1.57 | 5.47 | 5.17 |
| | | | | | 100 | 6.49 | 3.39 [a] | 2.37 | 2.02 | 2.60 | 2.26 | 2.77 | 2.30 | 3.06 | 2.25 | 10.80 | 8.03 |
| 80 | 5.64 | 1.75 | 6.86 | 2.17 [a] | 20 | 3.82 | 2.49 [a] | 2.52 | 1.54 | 2.03 | 1.79 | 1.79 | 1.67 | 1.67 | 1.96 | 8.00 | 5.47 |
| | | | | | 100 | 7.94 | 2.60 [c] | 2.94 | 2.01 | 3.52 | 1.87 [b] | 3.58 | 1.77 | 3.52 | 1.95 | 13.55 | 7.05 |
| 100 | 5.85 | 1.62 | 7.97 | 2.44 [a] | 20 | 3.03 | 2.53 [a] | 2.76 | 1.89 | 2.79 | 1.85 | 2.42 | 2.03 | 2.79 | 2.18 | 10.76 | 5.85 |
| F-values | 1.00 | | 45.68*** | | 20 | .79 | | 2.66, p = .075 | | 6.79** | | 5.60** | | 5.40** | | 7.98*** | |
| Post-hoc | 20 = 80 = 100 | | 20 < 80 = 100 | | | 20 = 80 = 100 | | 20 = 80 = 100 20 < 100+ | | 20 = 80 = 100 20 < 100 | | 20 = 80 = 100 20 < 100 | | 20 = 80 < 100+ 20 < 100 | | 20 = 80 = 100 20 < 100 | |
| | | | | | 100 | 3.91, p = .05 | | 1.35 | | 3.28, p .075 | | 2.59 | | .80 | | 2.23 | |

Trust 1 = Trust at Baseline; Trust 2 = Trust after the Learning phase; Trust 2 = Trust after the test phase. Paired samples *t*-tests within condition comparing adjacent measurement points:

[a] p < .001,

[b] p < .01,

[c] p < .05

*** p < .001;

** p < .01;

+ p < .1;

For the 20% contingent test phase, no learning phase manipulation effects reached full significance (100 vs 20: mean difference = 0.90, 95% CI = [-0.09, 1.88], p = .087, Cohen's *d* = 0.51; 80 vs 20: mean difference = 0.65, 95% CI = [-0.33, 1.64], p = .327, Cohen's *d* = 0.42; 100 vs 80: mean difference = 0.24, 95% CI = [-0.76, 1.25], p = 1.00, Cohen's *d* = 0.14). This again provides some indication that prior contingency-related learning experiences might have an immediate impact on help seeking behavior. However, like in Study 2, the results seem less pronounced compared to what we found with the adult participants in Study 1.

**Does change in contingency affect trust and help seeking behavior?.** First, with regard to trust, for the 100% contingent test phase children, we conducted a one-way ANOVA with condition (20%, or 80% learning phase contingency) as independent variable and trust after the test phase as dependent variable, followed by Bonferroni post-hoc tests. Table 3 shows that, after the 100% contingent test phase, the initial learning phase manipulation effect did not reach significance (80 vs 20: mean difference = -1.45, 95% CI = [-2.92, 0.02], p = .052, Cohen's *d* = 0.48). Moreover, we conducted paired samples t-tests within each learning phase condition to test whether trust changed over time. During the 100% contingent test phase, trust increased significantly for both the 20% and 80% contingent learning phase condition children (20% learning phase: mean difference = -2.91, 95% CI = [-4.39, -1.44], t(34) = -4.02, p < .001, Cohen's *d* = -0.95; 80% learning phase: mean difference = -1.30, 95% CI = [-2.43, -0.17], t(32) = -2.35, p < .05, Cohen's *d* = -0.53; see also Table 3). This again seems to suggest that trust caught up after exposure to more positive experiences, but that more negative prior learning experiences persistently reduced trust levels.

For the 20% contingent test phase condition children, we conducted a one-way ANOVA with condition (20%, 80%, or 100% learning phase contingency) as independent variable and trust after the test phase as dependent variable, followed by Bonferroni post-hoc tests. Table 3 shows that, after the 20% contingent test phase, the learning phase effects between the conditions were fully deleted. Moreover, we conducted paired samples t-tests within each learning

phase condition to test whether trust changed over time. During the 20% contingent test phase, both the 80% and 100% contingent learning phase condition children sifnificantly lost trust (80% learning phase: mean difference = 3.27, 95% CI = [2.08, 4.47], $t(32)$ = 5.59, $p < .001$, Cohen's $d$ = 1.43; 100% learning phase: mean difference = 4.94, 95% CI = [3.50, 6.38], $t(32)$ = 7.00, $p < .001$, Cohen's $d$ = 1.99). These results suggest that negative care-related experiences in our experiment outweighed prior positive learning experiences.

Second, with regard to help seeking behavior, for the 100% contingent test phase children, a one-way ANOVA with learning phase condition (20% versus 80% learning phase condition) as independent variable and calling behavior in each of the three last blocks of test trials and total calling behavior as dependent variables, revealed no effects that reached significance. Paired-samples t-tests showed that the 80% contingent learning phase children showed a significant increase in help-seeking behavior comparing the first with the second block of test trials (80% learning phase: block 1 to 2: mean difference = -0.58, 95% CI = [-0.98, -0.17], $t(32)$ = -2.89, $p = .007$, Cohen's $d$ = -0.30). Furthermore, the difference between the 20% and 80% contingent learning phase children got fully erased in the 100% test phase condition.

For the 20% contingent test phase children, Table 3 shows that the 20% contingent learning phase children continued to call significantly less for help compared to the 100% contingent learning phase children (block 2: mean difference = 1.53, 95% CI = [0.52, 2.55], $p < .01$, Cohen's $d$ = 0.90; block 3: mean difference = 1.40, 95% CI = [0.38, 2.42], $p < .01$, Cohen's $d$ = 0.78; block 4: mean difference = 1.45, 95% CI = [0.33, 2.58], $p < .01$, Cohen's $d$ = 0.77; total: mean difference = 5.29, 95% CI = [2.06, 8.51], $p < .001$, Cohen's $d$ = 0.96). In addition, the 20% contingent learning phase children's help seeking behavior further dropped significantly from the first to the second block of trials (20% learning phase: block 1 to 2: mean difference = -0.61, 95% CI = [0.28, 0.94], $t(35)$ = 3.80, $p < .001$, Cohen's $d$ = 0.39). This finding is in line with Ainsworth's observation that children who start to avoid seeking parental support are those children whose parents are consistently absent in their care [26].

## General discussion

With three independent studies, we aimed to experimentally examine whether trust development can be explored from a conditioning or expectancy-learning perspective and whether trust develops conditional upon contingency in terms of likelihood that a caregiver (Conditional Stimulus, CS) is associated with successful help with a problem (Unconditional Stimulus, UCS). We aimed to investigate whether this theoretical framework could be informative to understand (1) whether higher contingency in terms of successful support by a caregiver leads to increased trust, and (2) to more help seeking behavior, and (3) whether change in contingency affects trust and help seeking behavior. Results were largely in line with the predictions, providing first proof of concept that trust development might be studied from a conditioning perspective. Moreover, at a more detailed level, results suggested that more experimental research on the effects of contingency on trust development could provide new insights in the developmental dynamics underlying trust-related fluctuations, and eventually regarding the development of attachment.

### Research Question 1: Does higher contingency lead to more trust?

Across all three studies, we found that level of contingency was related to level of trust. Children who were exposed to a support figure that was more likely to provide successful help developed more trust in that support figure. This finding is in line with the existing broadband research showing that more sensitive and responsive parenting is linked with elevated levels of trust in parental support [5], but adds to this research because it provides proof of concept that

contingency could at least partly explain the causal mechanism underlying this link and because it suggests that it might be important to account for the relative contribution of single learning experiences across trust development. An important limitation of the current paradigm is that the distress in the task is mild, and that help seeking to solve a distressing problem reflects only a part of the rich complexity of attachment behavior and attachment relationships. In other words, the ecological validity of the findings should be further evaluated. In favor of the paradigm's relevance was the finding that baseline trust in the novel supportive figure was slightly but significantly correlated to trust in the mother in Study 3. However, we need to stay cautious interpreting this correlation, because the effect size in Study 3 was small and because the effect could not be fully replicated in Study 2. So, it is not unlikely that we have not measured exactly the same process as it occurs in real-life attachment relationships. Nevertheless, the robustness of the majority of our findings suggests that more research and similar research with more ecologically valid designs might be critical to further advance attachment theory. In spite of these concerns, one experiment in adults has shown that manipulating the likelihood that an unknown supporter helps avoiding exposure to shocks affects participants' attentional processing and attachment-related appraisals of the supporter [37]. Although effects on trust were not measured in this study, it seems reasonable to assume that the manipulation could have affected trust as well. In all, the current studies do suggest that it could be theoretical relevant to conduct more research with these kind of paradigms to understand under which conditions trust develops.

### Research Question 2: Does higher contingency lead to more help seeking behavior?

Also across all three studies, we found that contingency was related to help seeking behavior. Children who were exposed to a support figure that was more likely to provide successful help were more likely to call that figure for help. Interestingly, this effect was most explicit in study 1 in which adults were tested. Instead, in studies 2 and 3, children mainly showed this effect when comparing extremes (20%/50% versus 100% learning phase condition children). One reason might be that adults rely on more extensive real-life learning experiences regarding whether or not they can trust a caregiver [38]. As a result, adults in our study might have been more motivated to protect themselves from new harm. Instead, children are much more susceptible to positive attachment experiences [39] and may have been more motivated to retry to seek help in spite of earlier negative learning experiences.

### Research Question 3: Does change in contingency affects trust and help seeking behavior?

The current proof of concept results suggest that studying contingency effects on trust development might be especially interesting to investigate the effect of changes in the quality of care over time. So far, such research could only be conducted in so-called naturalistic experiments like adoption [40]. However, adoption remains rare and the impact of adoption is hard to distinguish from other factors related to child characteristics, pre-adoption experiences, and post-adoption experiences. Like in adoption research, the current findings obtained with our paradigm do suggest substantial catch-up in trust development and support seeking if adverse learning circumstances are followed by more beneficial circumstances [41]. The results also suggest that the initial learning effects continued to have an effect even after a more positive learning experience. This raises the hypothesis that prior more negative experiences have a lasting negative impact on trust. Instead, more positive learning experiences seemed quite sensitive to negative changes in the quality of the caregiver's response. Interestingly, these changes

seemed to have had a stronger effect at the level of expectations than at the level of behavior. Real life attachment research could test whether, indeed, children are inclined to continue seeking support in spite of a drop of trust. This illustrates how our hypothesis and paradigms like the one we used in the current studies to yield proof of concept could lead to novel developmental hypotheses that can be tested more directly with more naturalistic research paradigms. It would be interesting to find evidence for the counter-intuitive finding that changes in trust are not immediately followed by changes in help seeking behavior.

## Limitations

Although these findings are promising and could shed new light on trust development and even attachment theory, more research is needed to establish the ecological validity of the paradigm's results. More specifically, at this point one could argue that we merely measured learning rates of reward instead of processes related to attachment development. In line with this concern, we cannot rule out the possibility that similar effects could have been found had we instructed the children they were playing with a computer. Thus, it would be premature to argue that the current study contradicts attachment theory's traditional view that attachment is an inborn mechanism that does not require learning processes [18]. A next step would be to investigate whehter these findings translate to more naturalistic research. With that regard, it is promising that the current pattern of findings does not contradict existing more broadband attachment research. Also, the paradigm allows more sophisticated manipulations that allow to better understand how trust develops. However, for now, we were mostly interested to evaluate the extent to which the findings converge with existing knowledge and to see whether the manipulations could improve the knowledge-base in this area of research. With that regard, across the three studies, we found important evidence in favor of this theoretical and research-related approach to the phenomenon of trust development.

In spite of these limitations, the current findings are theoretically relevant. In three studies, we found evidence that trust development, and related support-seeking behavior, might be understood from learning theories regarding classical and operant conditioning. More specifically, results suggest that trust might develop as an expectancy-learning process with contingency (likelihood that help is successful) as a fundamental mechanism explaining differences in trust and support seeking. This finding is in line with the increasing awareness that trust development underlies a cognitive learning process resulting in the development of a secure base script [9]. This insight could help improve theory about trust development and could help shape interventions aimed at restoring trust in order to remedy the development of emotional and behavioral problems. Through this approach, we might be able to identify for individual children the level of contingency they need to increase their trust in the availability of their caregiver. With such information, it becomes possible to design interventions that promote trust development. For example, we could start creating the circumstances in which contingency of the caregiver is more guaranteed.

## Supporting information

**S1 Data. Dataset Study 1.**
(SAV)

**S2 Data. Dataset Study 2.**
(SAV)

**S3 Data. Dataset Study 3.**
(SAV)

## Author Contributions

**Conceptualization:** Guy Bosmans, Theodore E. A. Waters, Dirk Hermans.

**Data curation:** Chloe Finet.

**Formal analysis:** Guy Bosmans, Chloe Finet, Dirk Hermans.

**Funding acquisition:** Guy Bosmans.

**Methodology:** Guy Bosmans, Theodore E. A. Waters, Simon De Winter, Dirk Hermans.

**Project administration:** Simon De Winter.

**Resources:** Guy Bosmans, Theodore E. A. Waters.

**Software:** Simon De Winter.

**Supervision:** Guy Bosmans, Dirk Hermans.

**Writing – original draft:** Guy Bosmans.

**Writing – review & editing:** Guy Bosmans, Theodore E. A. Waters, Chloe Finet, Simon De Winter, Dirk Hermans.

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
