## [Decision Letter · Decision Letter 0]

12 Jul 2019

PONE-D-19-16003

Trust Development as an Expectancy-learning Process Testing Contingency Effects

PLOS ONE

Dear Dr. Bosmans,

Thank you for submitting your manuscript to PLOS ONE. After careful consideration, we feel that it has merit but does not fully meet PLOS ONE’s publication criteria as it currently stands. Therefore, we invite you to submit a revised version of the manuscript that addresses the points raised during the review process.

I was able to receive the input of two reviewers for this manuscript and I thank them for their careful attention to their work. I have also read the manuscript carefully and, like the reviewers, have mixed feelings about the work. The experimental paradigm is quite strong. The work is moving forward on the hypothesis that learning contingencies influences the development, emergence, or changes in trust. This kind of learning work has been becoming more frequently studied with regard to human computer interaction (e.g., https://www.ncbi.nlm.nih.gov/pubmed/22046724). Reviewer 1 identifies several related lines of work that were not described in the manuscript.

The results also challenge the framing of the work within an attachment context. Based on the paradigm used and the very modest effect sizes noted between the attachment measure and behaviorally assessed trust, this framing is very tenuous. Reviewer 2 identifies several instances where interpretations along these lines goes beyond the data.

Both Reviewers offered comments about the magnitude of effects and I will press on them further. The correlation estimates are very modest and, based on the sample sizes, the studies were underpowered to detect these associations. There are other analytic decisions that are challenging to reconcile. Given the study design, it was not clear why analyses did primarily rely on the RM-ANOVAs and the post-hoc tests from that model. There were multiple t-tests conducted that would be enveloped within the larger RM-ANOVA model.

Because of these conceptual and empirical challenges, the conclusions go beyond what the data indicate. Thus, I cannot recommend a favorable disposition for your manuscript at this time. I offer you the opportunity to significantly revise your manuscript that I know will be very effortful.

We would appreciate receiving your revised manuscript by Aug 26 2019 11:59PM. To enhance the reproducibility of your results, we recommend that if applicable you deposit your laboratory protocols in protocols.io, where a protocol can be assigned its own identifier (DOI) such that it can be cited independently in the future. For instructions see: http://journals.plos.org/plosone/s/submission-guidelines#loc-laboratory-protocols

We look forward to receiving your revised manuscript.

Kind regards,

Thomas M. Olino

Academic Editor

PLOS ONE

Journal Requirements:

2. We note that you have indicated that data from this study are available upon request. PLOS only allowsdata to be available upon request if there are legal or ethical restrictions on sharing data publicly. For more information on unacceptable data access restrictions, please see http://journals.plos.org/plosone/s/data-availability#loc-unacceptable-data-access-restrictions.

3. We note that on page 8 it is stated that the screen thanked children for feeding the donkey, yet participants in this part of the study were adults. Please correct and clarify.

Please note that according to our submission guidelines (http://journals.plos.org/plosone/s/submission-guidelines), outmoded terms and potentially stigmatizing labels should be changed to more current, acceptable terminology. For example: “Caucasian” should be changed to “white” or “of [Western] European descent” (as appropriate).

4. Please provide additional details regarding participant consent. Please ensure that the description of each of the experiments performed includes a description of the consent procedures. In the Methods section(s), please ensure that you have specified (1) whether consent was informed and (2) what type you obtained (for instance, written or verbal). Please state whether you obtained consent from parents or guardians for all studies including minors.

5. Please amend either the title on the online submission form (via Edit Submission) or the title in the manuscript so that they are identical.

6. Please include a title for tables 1-3.

Reviewers' comments:

Reviewer's Responses to Questions

**Comments to the Author**

1. Is the manuscript technically sound, and do the data support the conclusions?

Reviewer #1: Yes

Reviewer #2: Partly

2. Has the statistical analysis been performed appropriately and rigorously? 

Reviewer #1: Yes

Reviewer #2: Yes

3. Have the authors made all data underlying the findings in their manuscript fully available?

Reviewer #1: Yes

Reviewer #2: Yes

4. Is the manuscript presented in an intelligible fashion and written in standard English?

Reviewer #1: Yes

Reviewer #2: Yes

5. Review Comments to the Author

Reviewer #1: Overall this is an excellent set of studies that build on each other and present an important methodological approach to applying learning theory to trust development. The data are convincing and the methods are sound. I have a few requests:

1) The authors should probably refrain from referring to "marginally significant" results as this is technically a violation of frequentist statistical approaches from most points of view, although this is an admittedly minor issue. 2)

2)More importantly, it would be useful for the authors to provide both confidence intervals on point estimates and effect size information. This type of information is critical for determining the soundness of the results and understanding the degree of uncertainty with which the effects are estimated.

The results are in line with previous findings and theory, but offer important advances in methodology and a first guess as to what "responsiveness" or "good enough" really means. This type of work is critical for advancing the field's knowledge regarding attachment learning and trust development. Overall this manuscript is very well written and provides compelling evidence for the theoretical ideas therein.

A final note, similar work in the adult attachment literature has found similar evidence for the same learning theory based approach. The authors may be interested in these articles given the similarity in theory and approach.

Beckes, L. Simons, K., Lewis, D., Le, A., & Edwards, W. L. (2017). Desperately seeking support: Negative reinforcement schedules in the formation of adult attachment associations. Social Psychological and Personality Science, 8, 229-238.

Beckes, L., & Coan, J. A. (2015). The distress-relief dynamic in attachment bonds. In C. Hazan & V. Zayas (Eds.), Bases of Adult Attachment: Linking Brain, Mind, and Behavior (pp. 11-33). New York, NY, US: Springer Science + Business Media.

Beckes, L., Coan, J. A., & Morris, J. P. (2013). Implicit conditioning of faces via the social regulation of emotion: ERP evidence of early attentional biases for security conditioned faces. Psychophysiology, 50, 734-742.

Beckes, L., Simpson, J. A, & Erickson, A. B. (2010). Of snakes and succor: Learning secure attachment associations with novel faces via negative stimulus pairings. Psychological Science, 21, 721 – 728.

Reviewer #2: The authors use a learning theory approach, seeking to illuminate the processes by which child-caregiver trust and help seeking develop. Using a task in which adults and children learn reward schedules from a "caregiver" the authors measured trust as a function of caregivers who are likely to provide the correct answer at different rates. The study is an interesting possible analogue to these processes and the learning theory account is welcome. I agree with the authors that such a framework is likely to yield specific hypotheses. At the same time, I have feedback the authors may consider:

1. One concern I have with the paper is that the authors say they are seeking to illuminate the "underlying developmental mechanism" of trust and help seeking with caregivers. However, their study can speak very little to these processes. It is unclear whether the results are consistent with reward learning in close relationships, or, alternatively, the authors would have found the exact same results if participants were told they were playing with a computer. The authors seek to capture a process, but only show that trust increases with certain rates of reward and not others, and help seeking is higher in some conditions than others. It is not clear, however, whether what they describe is consistent with the processes of developing trust or seeking help in attachment contexts. Just as much as these data could describe a process somewhat similar to developing trust in close relationships, it could just as easily describe adults and children learning reward rates in a laboratory task.

2. Similarly, it is possible that learning rates of reward explains these developmental processes, but this ignores the extensive evidence of an attachment system that predisposes children to maintain proximity and use caregivers as a source soothing. I wonder how the authors might reconcile findings from attachment theory with their own approach?

3. There are times when the discussion (which appears frequently in the results section) seems to get too far from the data and the limitations of the approach. The following two sentences are examples, but not the only instances:

"So, this was the first evidence suggesting that good enough mothering requires at least 80% contingent care”.

"These results suggest that negative care-related experiences outweigh prior positive learning experiences.”

The framework is simply not strong enough to support these claims.

Similarly, the authors found that trust in one's caregiver was related to baseline trust in a "novel caregiver", though the correlations were low (r = .24 and r = .16) with only one truly significant. These data are not overwhelmingly convincing - I think if we posited the hypothesis that trust within these two contexts was different, we'd find ourselves with some evidence to suggest that they are. Could the authors provide a more constrained reporting of the meaning of these results?

4. The authors start the paper seeking to justify their experimental approach. While I understand their desire to point out the limitations of previous research, I wonder if they may go too far with sentences like “Clinically, broadband correlational studies are difficult to translate into concrete therapeutic strategies.” I’m not sure that they are, and there are treatments that have been developed based on this research. The authors may want to find another way to introduce their study, particularly because the major limitation of the current study is that ecological validity is decreased by the experimental approach with “new caregivers”, which may make it quite difficult to translate these findings into the clinical realm.

6. PLOS authors have the option to publish the peer review history of their article (what does this mean?). If published, this will include your full peer review and any attached files.

Reviewer #1: No

Reviewer #2: No

---

## [Author Response · Author response to Decision Letter 0]

6 Sep 2019

Responses to the Editor

We have completely reworked the document to comply with PLOS One's style requirements. 

2. We note that you have indicated that data from this study are available upon request. PLOS only allowsdata to be available upon request if there are legal or ethical restrictions on sharing data publicly. 

We have prepared the files and uploaded them as Supporting Information Files. 

3. We note that on page 8 it is stated that the screen thanked children for feeding the donkey, yet participants in this part of the study were adults. Please correct and clarify.

Please note that according to our submission guidelines outmoded terms and potentially stigmatizing labels should be changed to more current, acceptable terminology. For example: “Caucasian” should be changed to “white” or “of [Western] European descent” (as appropriate).

This phrasing was altered as requested. 

4. Please provide additional details regarding participant consent. Please ensure that the description of each of the experiments performed includes a description of the consent procedures. In the Methods section(s), please ensure that you have specified (1) whether consent was informed and (2) what type you obtained (for instance, written or verbal). Please state whether you obtained consent from parents or guardians for all studies including minors.

In study 1, adults' written consent was obtained. For both Studies 2 & 3, parent and child written consent was obtained.

5. Please amend either the title on the online submission form (via Edit Submission) or the title in the manuscript so that they are identical.

This was amended as requested

6. Please include a title for tables 1-3.

All tables received titles 

Responses to the Reviewers

Response to Reviewer #1:

Overall this is an excellent set of studies that build on each other and present an important methodological approach to applying learning theory to trust development. The data are convincing and the methods are sound. I have a few requests:

We are thankful for this positive evaluation of our paper and grateful for the helpful suggestions thanks to which the paper was substantially strengthened. 

1) The authors should probably refrain from referring to "marginally significant" results as this is technically a violation of frequentist statistical approaches from most points of view, although this is an admittedly minor issue.

We agree that it is wiser to be more cautious about valuing marginally significant effects. We thought it was most prudent to remove any discussion of marginally significant effects. Moreover, we no longer refer to marginally significant effects. Instead, where appropriate or necessary for the flow of the story we now more carefully describe that effects did not reach significance. 

2)More importantly, it would be useful for the authors to provide both confidence intervals on point estimates and effect size information. This type of information is critical for determining the soundness of the results and understanding the degree of uncertainty with which the effects are estimated.

Following this request, we gave effect size information wherever possible. Moreover, we additionally provided confidence intervals for point estimates. Given the amount of analyses we could discuss, we wanted to keep the results section concise, so for now we decided to provide all the information for all the significant effects and for the non-significant effects that were important to make a theoretical point. 

3) The results are in line with previous findings and theory, but offer important advances in methodology and a first guess as to what "responsiveness" or "good enough" really means. This type of work is critical for advancing the field's knowledge regarding attachment learning and trust development. Overall this manuscript is very well written and provides compelling evidence for the theoretical ideas therein.

We are very grateful for these words of praise! 

4) A final note, similar work in the adult attachment literature has found similar evidence for the same learning theory based approach. The authors may be interested in these articles given the similarity in theory and approach.

Beckes, L. Simons, K., Lewis, D., Le, A., & Edwards, W. L. (2017). Desperately seeking support: Negative reinforcement schedules in the formation of adult attachment associations. Social Psychological and Personality Science, 8, 229-238.

Beckes, L., & Coan, J. A. (2015). The distress-relief dynamic in attachment bonds. In C. Hazan & V. Zayas (Eds.), Bases of Adult Attachment: Linking Brain, Mind, and Behavior (pp. 11-33). New York, NY, US: Springer Science + Business Media.

Beckes, L., Coan, J. A., & Morris, J. P. (2013). Implicit conditioning of faces via the social regulation of emotion: ERP evidence of early attentional biases for security conditioned faces. Psychophysiology, 50, 734-742.

Beckes, L., Simpson, J. A, & Erickson, A. B. (2010). Of snakes and succor: Learning secure attachment associations with novel faces via negative stimulus pairings. Psychological Science, 21, 721 – 728.

We are very grateful for this suggestion. We do know the work of Lane Beckes and highly appreciate his papers. Because we agree that mentioning this work strenghtenes the paper, we followed this suggestion. We discussed his work at two places in the paper. 

First, in the introduction, we were asked by reviewer 2 (comments 1 & 2) to contrast our learning theory model of attachment development with attachment theory's traditional criticism on a learning theoretical approach to attachment. In the flow of this argumentation in the introduction, we now mention these papers to make the point that support has been found for such mechanisms in adult attachment research: 

p4: More recently, researchers started to argue that this biological preparedness of infants to establish attachment relationships with caregivers does not fully explain why individual differences in children's expectations or trust about caregivers' availability for support develop [8]. One thusfar understudied possibility could be that learning models might be a helpful addition to attachment theory to explain these differences [19]. Recent literature argues in a similar way that adult attachment development reflects a conditioning process [20-22].

Second, in the discussion, we added critique on our paradigm, as we still lack convincing evidence that trust towards the avatar is attachment relevant. Although the Beckes et al., 2017 study did not directly target trust, the theoretical assumption and operationalization is very comparable and the results are also convergent with our studies. So we mentioned the latter paper to strengthen our argument that in spite of the validity of these concerns, our paradigm could still yield useful results. 

p27: This all suggests that more research and similar research with more ecologically valid designs is crucial in the future, In spite of these concerns, one experiment in adults has shown that manipulating the likelihood that an unknown supporter helps avoiding exposure to shocks affects participants' attentional processing and attachment-related appraisals of the supporter [36]. Although effects on trust were not measured in this study, it seems reasonable to assume that the manipulation could have affected trust as well.

Responses to Reviewer #2:

The authors use a learning theory approach, seeking to illuminate the processes by which child-caregiver trust and help seeking develop. Using a task in which adults and children learn reward schedules from a "caregiver" the authors measured trust as a function of caregivers who are likely to provide the correct answer at different rates. The study is an interesting possible analogue to these processes and the learning theory account is welcome. I agree with the authors that such a framework is likely to yield specific hypotheses. At the same time, I have feedback the authors may consider:

Thank you for your generous feedback and appreciation of the paper. Using the feedback while revising the paper has significantly improved the manuscript. 

1. One concern I have with the paper is that the authors say they are seeking to illuminate the "underlying developmental mechanism" of trust and help seeking with caregivers. However, their study can speak very little to these processes. It is unclear whether the results are consistent with reward learning in close relationships, or, alternatively, the authors would have found the exact same results if participants were told they were playing with a computer. The authors seek to capture a process, but only show that trust increases with certain rates of reward and not others, and help seeking is higher in some conditions than others. It is not clear, however, whether what they describe is consistent with the processes of developing trust or seeking help in attachment contexts. Just as much as these data could describe a process somewhat similar to developing trust in close relationships, it could just as easily describe adults and children learning reward rates in a laboratory task.

We agree that the current design, in spite of its interesting characteristics cannot rule out this alternative interpretation. Therefore, we devoted substantial attention to this issue in the limitations section. We now wrote: 

p29: More specifically, at this point one could argue that we merely measured learning rates of reward instead of processes related to attachment development. In line with this concern, we cannot rule out the possibility that similar effects could have been found had we instructed the children they were playing with a computer. Thus, it would be premature to argue that the current study contradicts attachment theory's traditional view that attachment is an inborn mechanism that does not require learning processes [18]. A next step would be to investigate whehter these findings translate to more naturalistic research. With that regard, it is promising that the current pattern of findings does not contradict existing more broadband attachment research.

2. Similarly, it is possible that learning rates of reward explains these developmental processes, but this ignores the extensive evidence of an attachment system that predisposes children to maintain proximity and use caregivers as a source soothing. I wonder how the authors might reconcile findings from attachment theory with their own approach?

We agree that this is an important issue that we did not explicitly address in the previous version of the paper. In fact, this refers back to a very old discussion between the more evolutionary view on attachment as proposed by Bowlby and Ainsworth and learning theory like, for example, proposed by for example Gerwitz. This discussion has been settled in a seminal paper from Rajecki, Lamb, and Obmascher (1978). Ever since, it has been accepted that attachment does not develop due to learning experiences. 

However, historically, Bowlby has never been fully opposed against the idea that learning processes could partly explain aspects of attachment development. He did not touch this issue in his published work, but he referred to this in his correspondence with colleagues. Thanks to Robbie Duschinsky, we got access to some of these letters and we mentioned them in the new version of the paper. 

Moreover, it should be noted that the argumentation of Rajecki et al. refers to why children develop an attachment bond to parents and less to why individual differences in attachment (in)security develop. Recent work increasingly argues that with regard to that issue, learning theory could prove to be an added value. We now describe this debate and this argumentation in the introduction. 

p4: 

 The idea that learning theory could explain at least a part of attachment development, has traditionally been focus of fierce debate [15]. When developing his attachment theory, Bowlby was in a difficult position. On the one hand he wanted to emphasise that attachment is an evolutionarily primed behavior system, not reducible to classical or operant conditioning [16]. On the other hand, he also held that the attachment behavioral system is assembled and elaborated in the context of learned experiences [17]. Nevertheless, he mostly left the topic alone. Later, attachment researchers mostly demonstrated that children develop their attachment relationships independent of the quality of care, interpreting this finding as evidence that the attachment behavioral system is an evolutionary driven system that requires no learning experiences to be established [15, 18]. 

More recently, researchers started to argue that this biological preparedness of infants to establish attachment relationships with caregivers does not fully explain why individual differences in children's expectations or trust about caregivers' availability for support develop [8]. One thusfar understudied possibility could be that learning models might be a helpful addition to attachment theory to explain these differences [19]. Recent literature argues in a similar way that adult attachment development reflects a conditioning process [20-22]. Focusing on childhood attachment, Bosmans [23] proposed that children learn to trust (CR) in a responsive primary caregiver like the mother (CS) when the mother gets associated with the repeated experience that she provides successful support during distress (UCS), which is automatically followed by a sense of relief and a sense of security (UCR). At the level of operant conditioning, it has been proposed that trust versus lack of trust in support during distress (Sd) increases or decreases the likelihood that children will seek support (R) during distress. More versus less support seeking is respectively reinforced by the fact that sensitive caregivers help to solve problems more easily or by the fact children avoid experiencing the anticipated negative effects of rejection or inadequate support [23].

In addition, we referred back to this discussion in the discussion section: 

p29: Thus, it would be premature to argue that the current study contradicts attachment theory's traditional view that attachment is an inborn mechanism that does not require learning processes [18]. A next step would be to investigate whehter these findings translate to more naturalistic research. With that regard, it is promising that the current pattern of findings does not contradict existing more broadband attachment research.

3. There are times when the discussion (which appears frequently in the results section) seems to get too far from the data and the limitations of the approach. The following two sentences are examples, but not the only instances:

Overall, we tried to be more careful in phrasing our interpretations. With regard to the specific examples, we made the following changes: 

"So, this was the first evidence suggesting that good enough mothering requires at least 80% contingent care”.

We now wrote:

"So, if this finding could be replicated in more naturalistic research on parent-child interactions, the finding could be used as an argument that good enough mothering might require at least 80% contingent care."

"These results suggest that negative care-related experiences outweigh prior positive learning experiences.”

We now wrote: 

"This again seems to suggest that trust catches up after exposure to more positive

experiences, but that more negative prior learning experiences persist reducing trust levels."

Similarly, the authors found that trust in one's caregiver was related to baseline trust in a "novel caregiver", though the correlations were low (r = .24 and r = .16) with only one truly significant. These data are not overwhelmingly convincing - I think if we posited the hypothesis that trust within these two contexts was different, we'd find ourselves with some evidence to suggest that they are. Could the authors provide a more constrained reporting of the meaning of these results?

We now wrote:

"However, in favor of the paradigm’s relevance was the finding that baseline trust in the novel supportive figure was slightly but significantly correlated to trust in the mother in Study 3. Nevertheless, we need to stay cautious interpreting this correlation, as it could not be fully replicated in Study 2. This might have been due to reduced power, but could also indicate that we created a less relevant experimental context. This all suggests that more research and similar research with more ecologically valid designs is crucial in the future,"

4. The authors start the paper seeking to justify their experimental approach. While I understand their desire to point out the limitations of previous research, I wonder if they may go too far with sentences like “Clinically, broadband correlational studies are difficult to translate into concrete therapeutic strategies.” I’m not sure that they are, and there are treatments that have been developed based on this research. The authors may want to find another way to introduce their study, particularly because the major limitation of the current study is that ecological validity is decreased by the experimental approach with “new caregivers”, which may make it quite difficult to translate these findings into the clinical realm.

We agree with this concern, and when rereading the paper, we felt that the setup of the study did not really require such a clinical argument. Consequently, we decided to drop this sentence from the introduction.

---

## [Decision Letter · Decision Letter 1]

29 Oct 2019

PONE-D-19-16003R1

Trust Development as an Expectancy-learning Process: 

Testing Contingency Effects.

PLOS ONE

Dear Dr. Bosmans,

Thank you for submitting your manuscript to PLOS ONE. After careful consideration, we feel that it has merit but does not fully meet PLOS ONE’s publication criteria as it currently stands. Therefore, we invite you to submit a revised version of the manuscript that addresses the points raised during the review process.

Please see the comments below from the Academic Editor.

We would appreciate receiving your revised manuscript by Dec 13 2019 11:59PM. To enhance the reproducibility of your results, we recommend that if applicable you deposit your laboratory protocols in protocols.io, where a protocol can be assigned its own identifier (DOI) such that it can be cited independently in the future. For instructions see: http://journals.plos.org/plosone/s/submission-guidelines#loc-laboratory-protocols

We look forward to receiving your revised manuscript.

Kind regards,

Thomas M. Olino

Academic Editor

PLOS ONE

Additional Editor Comments (if provided):

I have received feedback from both of the original reviewers. You will see that Reviewer 1 is fully satisfied with the revision, but Reviewer 2 continues to have some significant reservations. I concur with the sentiment that the work of integrating learning into attachment research is important. At the same time, there are comments in the manuscript that go beyond the data. Reviewer 2 is very specific in the types of comments for which this is a significant concern.

Please also carefully review the percentage of contingencies through the sections on Study 3. There seem to be inaccuracies in 50 vs. 70 vs. 100 compared to the test contingencies of 20 vs. 80. vs. 100.

Reviewers' comments:

Reviewer's Responses to Questions

**Comments to the Author**

1. If the authors have adequately addressed your comments raised in a previous round of review and you feel that this manuscript is now acceptable for publication, you may indicate that here to bypass the “Comments to the Author” section, enter your conflict of interest statement in the “Confidential to Editor” section, and submit your "Accept" recommendation.

Reviewer #1: All comments have been addressed

Reviewer #2: (No Response)

2. Is the manuscript technically sound, and do the data support the conclusions?

Reviewer #1: Yes

Reviewer #2: Partly

3. Has the statistical analysis been performed appropriately and rigorously? 

Reviewer #1: Yes

Reviewer #2: Yes

4. Have the authors made all data underlying the findings in their manuscript fully available?

Reviewer #1: Yes

Reviewer #2: Yes

5. Is the manuscript presented in an intelligible fashion and written in standard English?

Reviewer #1: Yes

Reviewer #2: Yes

6. Review Comments to the Author

Reviewer #1: Overall I think this is a very important contribution to the attachment literature. Attachment theory needs to be integrated with learning theory. Whereas Bowlby created a compelling overall model of attachment and how it develops, given the paucity of data on neurobiology at the time it inadequately addresses the fundamental mechanisms underlying attachment learning. Although one may argue that the underlying mechanism of attachment is an innate behavioral system, and therefore no standard learning mechanisms can explain attachment dynamics, I believe this is a fundamental error in both reading Bowlby and for the future of attachment theory writ large. Bowlby argues for a goal corrected behavioral system. This implies learning. The mechanisms he uses to explain such learning are based in a perspective of homeostasis that no longer comports with our understanding of physiological regulation. It is clear that learning processes are subserved by general purpose neural systems that function in a manner well characterized by learning theory, and that those systems are critical for the regulation of physiological and psychological needs. Bowlby was working in a time when behaviorists were too narrow minded about the breadth of basic rewards and punishments. They completely ignored the possibility that social contact was a very real need in the same vein as food and water. Moreover, their blank slate view of all animals was unsustainable. As such theorists like Bowlby filled that vacuum with models that did not rely on learning theory based mechanisms. Unfortunately for the field it has taken far too long to reconcile and integrate Bowlby's critical contributions with those of the behaviorists and cognitive learning theorists that followed. This is an important step in that direction and I applaud the authors for their creativity and boldness in furthering this integration.

Reviewer #2: The authors have made some major revisions to their manuscript which I think improve the overall product. At the same time, I still think their conclusions go well beyond the bounds of their research design in places and could be further qualified. I have specific feedback below:

Point 2. The authors make a valiant effort to discuss and resolve the tension between the role of learning theory in attachment and attachment as an innate biological system. I personally think there is a distinction between the processes of the attachment system, which seem to be innate, and attachment styles that are likely (at least partly) shaped by learning. It may be worthwhile to differentiate their discussion in this way, as they have started in their added text, to some degree. Attachment theorists often propose a model akin to learning theory to explain the development of anxious, avoidant or secure attachment styles, while at the same time noting Bowlby’s regard for attachment processes (meaning proximity seeking, secure base, etc.) as primary needs. Researchers often discuss, for instance, that secondary attachment styles are still ways to maintain attachment (e.g., proximity to caregiver) in non-ideal contexts (e.g., parent is rejecting of attachment needs). Delineating what is/can be learned may be more innate is an important discussion for this paper.

3. I still feel like these phrases are too much for the research method to support. There may quite a bit different about interacting with a real attachment figure during a critical timepoint in life that makes a lower percentage of contingent care necessary, for instance. I would advice deleting this claim, and all others like it. The authors show a proof of concept, which is interesting, but I would not try to push for the idea that we are really seeing attachment-like processes unfolding. The data don't seem to support that either (see 4).

4. The authors now note that their lack of a significant finding of a correlation between trust in a “novel caregiver” and trust in the mother may be due to reduced power. However, the effect size is simply extremely small in the case of study 2. Even if this were significant, given the author’s claims that they are measuring attachment processes is problematic when there is such a small relationship between these measures. I would omit the mention of reduced power being the fault and more clearly state that these data may argue against the same process being measured in both instances.

7. PLOS authors have the option to publish the peer review history of their article (what does this mean?). If published, this will include your full peer review and any attached files.

Reviewer #1: No

Reviewer #2: No

---

## [Author Response · Author response to Decision Letter 1]

6 Nov 2019

Additional Editor Comments:

Please also carefully review the percentage of contingencies through the sections on Study 3. There seem to be inaccuracies in 50 vs. 70 vs. 100 compared to the test contingencies of 20 vs. 80. vs. 100.

We are grateful that you noticed this error and changed this in the results section of Study 3. 

Reviewer #1: 

Overall I think this is a very important contribution to the attachment literature. Attachment theory needs to be integrated with learning theory. Whereas Bowlby created a compelling overall model of attachment and how it develops, given the paucity of data on neurobiology at the time it inadequately addresses the fundamental mechanisms underlying attachment learning. Although one may argue that the underlying mechanism of attachment is an innate behavioral system, and therefore no standard learning mechanisms can explain attachment dynamics, I believe this is a fundamental error in both reading Bowlby and for the future of attachment theory writ large. Bowlby argues for a goal corrected behavioral system. This implies learning. The mechanisms he uses to explain such learning are based in a perspective of homeostasis that no longer comports with our understanding of physiological regulation. It is clear that learning processes are subserved by general purpose neural systems that function in a manner well characterized by learning theory, and that those systems are critical for the regulation of physiological and psychological needs. Bowlby was working in a time when behaviorists were too narrow minded about the breadth of basic rewards and punishments. They completely ignored the possibility that social contact was a very real need in the same vein as food and water. Moreover, their blank slate view of all animals was unsustainable. As such theorists like Bowlby filled that vacuum with models that did not rely on learning theory based mechanisms. Unfortunately for the field it has taken far too long to reconcile and integrate Bowlby's critical contributions with those of the behaviorists and cognitive learning theorists that followed. This is an important step in that direction and I applaud the authors for their creativity and boldness in furthering this integration.

We were very grateful for these words of praise, as they express exactly what we hoped to achieve through this study. 

Reviewer #2: 

The authors have made some major revisions to their manuscript which I think improve the overall product. At the same time, I still think their conclusions go well beyond the bounds of their research design in places and could be further qualified. I have specific feedback below:

We are happy that the reviewer deems our changes in line with the majority of his/her prior requests. We agree that the additional suggestions indeed were necessary to allow readers to more adequately evaluate the contribution of the studies to the literature. 

Point 2. The authors make a valiant effort to discuss and resolve the tension between the role of learning theory in attachment and attachment as an innate biological system. I personally think there is a distinction between the processes of the attachment system, which seem to be innate, and attachment styles that are likely (at least partly) shaped by learning. It may be worthwhile to differentiate their discussion in this way, as they have started in their added text, to some degree. Attachment theorists often propose a model akin to learning theory to explain the development of anxious, avoidant or secure attachment styles, while at the same time noting Bowlby’s regard for attachment processes (meaning proximity seeking, secure base, etc.) as primary needs. Researchers often discuss, for instance, that secondary attachment styles are still ways to maintain attachment (e.g., proximity to caregiver) in non-ideal contexts (e.g., parent is rejecting of attachment needs). Delineating what is/can be learned may be more innate is an important discussion for this paper.

We are grateful that the reviewer was in agreement with the main points of the adjustment we made in response to the prior request. We do agree that the added suggestions are in line with the message that we wanted to communicate, so we adjusted these parts of the introduction so they would be more explicit in making the point that above an innate system the development of different attachment styles might reflect at least partly learning processes. We adjusted this part as follows (we underlined the changes we made):

"More recently, researchers started to argue that this biological preparedness of infants to establish attachment relationships with caregivers does not fully explain why individual differences in children's expectations or trust about caregivers' availability for support develop [8]. One thus far understudied possibility could be that learning models might be a helpful addition to attachment theory to explain attachment-related differences that are not innate [19], such as secondary (anxious, avoidant, and secure) attachment styles (Mikulincer & Shaver, 2017). Recent literature argues in a similar way that adult attachment development reflects a conditioning process [20-22]. Focusing on childhood attachment, Bosmans [23] proposed that children learn to trust (CR) in a responsive primary caregiver like the mother (CS) when the mother gets associated with the repeated experience that she provides successful support during distress (UCS), which is automatically followed by a sense of relief and a sense of security (UCR). At the level of operant conditioning, it has been proposed that trust versus lack of trust in support during distress (Sd) increases or decreases the likelihood that children will seek support (R) during distress. More versus less support seeking is respectively reinforced by the fact that sensitive caregivers help to solve problems more easily or by the fact children avoid experiencing the anticipated negative effects of rejection or inadequate support [23]. In sum, the current study builds on the idea that the processes of the attachment system are innate, but that attachment styles are (at least partly) shaped by learning." 

3. I still feel like these phrases are too much for the research method to support. There may quite a bit different about interacting with a real attachment figure during a critical timepoint in life that makes a lower percentage of contingent care necessary, for instance. I would advice deleting this claim, and all others like it. The authors show a proof of concept, which is interesting, but I would not try to push for the idea that we are really seeing attachment-like processes unfolding. The data don't seem to support tat either (see 4).

We agree that we could be even more cautious in positioning the studies within the broader area of attachment research and in interpreting the findings. Therefore, from the introduction onwards, we introduce the current studies as testing a proof of concept explaining why we opted to use this paradigm (focusing on a novel attachment figure and specifically on help provided by this figure). In the results sections, we avoided using language that suggests that we can extend our interpretations beyond what we found with this paradigm. Finally, in the discussion, we more carefully point again at the fact that this is a proof of concept study and that real-life attachment interactions might show different dynamics. Now we only make the point that the findings suggest that it could be useful to study contingency-related dynamics in attachment-relevant interactions with real attachment figures. 

In the introduction, we now wrote: 

(p6): "The current study aimed to provide proof of concept for the idea that contingency could explain part of the variance in the development of trust (Research Question 1) and in the development of support seeking (Research Question 2) and for the idea that questions regarding the definition of good enough mothering and regarding the dynamics of (in)stability of trust over time can in theory be explained by contingency-related effects (Research Question 3)."

(p7): " Although this approach will not straightforwardly translate to the complex interactions with a real attachment figure during critical timepoints in individuals' lives, the approach has the advantage that we can identify the role of contingency controlling for the impact of prior learning experiences with those real attachment figures. This way, we can experimentally test whether differences in contingency affect trust and test how changes in contingency from the learning to the test phase are linked to changes in trust. Finally, we can measure participants’ inclination to seek the caregiver’s help during the test phase, as a proxy of participants’ support seeking behavior."

In the results section of Study 1 we now wrote: 

(p. 13): "This suggests in adult participants that initial contingency effects had a lasting influence in trust development, even after exposure to a novel, improved experience."

In the results section of Study 2 we now wrote: 

(p. 16): " Like in Study 1, this suggests that trust development depended on the contingency of success in provided care and that trust development in the current paradigm required more than 70% contingent caregivers."

(p. 19): " Like in Study 1, this suggests that improving contingency increased trust, but that prior less contingent care experiences delayed this catch-up."

In the introduction to study 3 we removed the sentence that we were looking for an indication of how much contingency is needed to receive good enough mother. The sentences now are written as follows: 

(p. 20): " Moreover, Study 2 did not reveal which percentage of contingency suffices to increase trust levels compared to baseline. Therefore, we included an 80% instead of a 70% contingent condition, to test whether 80% contingency suffices to significantly increase trust."

In the results section of Study 3 we now wrote: 

(p. 21): " So, this suggests for the current experimental procedure that contingency needs to be at least 80% to be good enough for trust in the novel caregiver to increase."

(p.24): " This again seems to suggest that trust caught up after exposure to more positive experiences, but that more negative prior learning experiences persistently reduced trust levels."

(p.24): " These results suggest that negative care-related experiences in our experiment outweighed prior positive learning experiences."

In the general discussion, we now wrote: 

(p. 26) " Results were largely in line with the predictions, providing first proof of concept that trust development might be studied from a conditioning perspective. Moreover, at a more detailed level, results suggested that more experimental research on the effects of contingency on trust development could provide new insights in the developmental dynamics underlying trust-related fluctuations, and eventually regarding the development of attachment."

(p. 26): " This finding is in line with the existing broadband research showing that more sensitive and responsive parenting is linked with elevated levels of trust in parental support [5], but adds to this research because it provides proof of concept that contingency could at least partly explain the causal mechanism underlying this link and because it suggests that it might be important to account for the relative contribution of single learning experiences across trust development. "

(p 28): " The current proof of concept results suggest that studying contingency effects on trust development might be especially interesting to investigate the effect of changes in the quality of care over time. So far, such research could only be conducted in so-called naturalistic experiments like adoption [39]. However, adoption remains rare and the impact of adoption is hard to distinguish from other factors related to child characteristics, pre-adoption experiences, and post-adoption experiences. Like in adoption research, the current findings obtained with our paradigm do suggest substantial catch-up in trust development and support seeking if adverse learning circumstances are followed by more beneficial circumstances [40]. The results also suggest that the initial learning effects continued to have an effect even after a more positive learning experience. This raises the hypothesis that prior more negative experiences have a lasting negative impact on trust. Instead, more positive learning experiences seemed quite sensitive to negative changes in the quality of the caregiver's response. Interestingly, these changes seemed to have had a stronger effect at the level of expectations than at the level of behavior. Real life attachment research could test whether, indeed, children are inclined to continue seeking support in spite of a drop of trust. This illustrates how our hypothesis and paradigms like the one we used in the current studies to yield proof of concept could lead to novel developmental hypotheses that can be tested more directly with more naturalistic research paradigms. "

(p. 29-30): " In spite of these limitations, the current findings are theoretically relevant. In three studies, we found evidence that trust development, and related support-seeking behavior, might be understood from learning theories regarding classical and operant conditioning. More specifically, results suggest that trust might develop as an expectancy-learning process with contingency (likelihood that help is successful) as a fundamental mechanism explaining differences in trust and support seeking. This finding is in line with the increasing awareness that trust development underlies a cognitive learning process resulting in the development of a secure base script [9]. This insight could help improve theory about trust development and could help shape interventions aimed at restoring trust in order to remedy the development of emotional and behavioral problems. Through this approach, we might be able to identify for individual children the level of contingency they need to increase their trust in the availability of their caregiver. With such information, it becomes possible to design interventions that promote trust development. For example, we could start creating the circumstances in which contingency of the caregiver is more guaranteed."

4. The authors now note that their lack of a significant finding of a correlation between trust in a “novel caregiver” and trust in the mother may be due to reduced power. However, the effect size is simply extremely small in the case of study 2. Even if this were significant, given the author’s claims that they are measuring attachment processes is problematic when there is such a small relationship between these measures. I would omit the mention of reduced power being the fault and more clearly state that these data may argue against the same process being measured in both instances.

In line with this request, we omitted the discussion of power, emphasized more the small effect sizes we found, and added the suggested message that interpretation of the current studies' results require caution. 

(p. 27): " However, we need to stay cautious interpreting this correlation, because the effect size in Study 3 was small and because the effect could not be fully replicated in Study 2. So, it is not unlikely that we have not measured exactly the same process as it occurs in real-life attachment relationships. Nevertheless, the robustness of the majority of our findings suggests that more research and similar research with more ecologically valid designs might be critical to further advance attachment theory. "

---

## [Editor Report · Decision Letter 2]

18 Nov 2019

Trust Development as an Expectancy-learning Process: Testing Contingency Effects.

PONE-D-19-16003R2

Dear Dr. Bosmans,

We are pleased to inform you that your manuscript has been judged scientifically suitable for publication and will be formally accepted for publication once it complies with all outstanding technical requirements.

With kind regards,

Thomas M. Olino

Academic Editor

PLOS ONE

Additional Editor Comments (optional):

Thank you for your careful attention to the comments of Reviewer 2. They provide a tempered evaluation of the results that provides promise for future work.
---

## [Editor Report · Acceptance letter]

2 Dec 2019

PONE-D-19-16003R2 

Trust Development as an Expectancy-learning Process: Testing Contingency Effects. 

Dear Dr. Bosmans:

I am pleased to inform you that your manuscript has been deemed suitable for publication in PLOS ONE. Congratulations! Your manuscript is now with our production department. 

With kind regards,

on behalf of

Dr. Thomas M. Olino 

Academic Editor

PLOS ONE